# The Relationships among Experience, Delight, and Recollection for Revisit Intention in Chocolate Exposition

**Hyeon Mo Jeon** [1] , **Se Ran Yoo** [2] **and Seon Hee Kim** [3,*]

1   Department of Hotel, Tourism, and Foodservice Management, Dongguk University-Gyeongju, Gyeongju 38066, Korea; jhm010@dongguk.ac.kr
2   Department of Foodservice Management, Kyung Hee University, Seoul 02447, Korea; sel97@khu.ac.kr
3   Department of Culinary and Foodservice Management, Sejong University, 209, Neungdong-ro, Gwangjin-gu, Seoul 05006, Korea
*   Correspondence: anes12054@naver.com; Tel.: +82-10-4473-8050

**Abstract:** This study examined the relationship among holistic experience and delight, recollection, and revisit intention, in visitors to a chocolate exposition, to confirm the factors influencing their revisit intention. To accomplish this objective, a sample of 280 visitors to the Salon du Chocolat was utilized. The findings revealed that the visitors' memorable on-site experiences had a positive impact on their sense of delight and recollection. These emotions were shown to positively influence their intention to revisit the exposition. In conclusion, the on-site experiences of sense, feel, think, act, and relate, were the most important determinants of revisit intention among visitors. The findings of this study meaningfully contribute to theory by applying the concepts of experiential marketing to food expositions. Our study also proposes practical suggestions for hosting food expositions in a post-COVID world. The theoretical framework proposed and tested for model validation lay the foundation for future research on experiential marketing of food expositions.

**Keywords:** chocolate exposition; experience; delight; recollection; revisit intention

---

## 1. Introduction

The global chocolate market is expected to register a compound annual growth rate of 5% from 2019 to 2023, amid continuous technological innovation in the industrial chocolate supply chain and increased consumer awareness of the benefits of dark chocolate [1]. Increased perception of the utilitarian value of chocolate coupled with the growth of luxury food consumption, is popularizing the concept of bean-to-bar chocolates (produced directly from cocoa beans) and specialty chocolate stores [2,3]. The chocolate market has become more segmented than ever, as chocolate is a preferred gift for yourself and other various age groups [4].

Food-oriented festivals and tourism are a new trend that has been actively adopted by the hospitality industry [5]. Visitors to wine and food festivals enjoy the themed, fun atmosphere, and experience a feeling of freedom, of escaping from their boring everyday lives [6]. The exhibitions of historical and cultural heritage have served as venues enabling social exchange and special experiences [7]. Entertainment, a consumption behavior that combines eating with entertainment, is an emerging consumer trend; the hospitality experience, which combines eating and leisure, is now widespread in the food industry [8]. The FICO Eataly World, an agri-food theme park in Italy, features nearly 30 different events and 50 lectures that are designed to provide customers with a variety of food and agricultural experiences [8,9]. In the chocolate industry, the Hershey Company operates five chocolate theme parks globally called Hershey's Chocolate World. These offer exciting chocolate

adventures, including real participation in the chocolate manufacturing process, chocolate-themed rides, and unique chocolate-themed shopping experiences [10].

In line with current trends in experiential consumption, expositions for chocolate products are being hosted by the dessert sector. Since its inception in France, in 1994, the Salon du Chocolat has been held annually in 32 cities in 16 countries, gaining attention as the world's premier chocolate show. The year 2020 marks its twenty-fifth anniversary. This event attracts local and international dessert enthusiasts, as well as culinary experts [11]. During the event, artisans from all over the world present their specialty chocolate products in a separate hall and provide visitors with an opportunity to discover and experience a unique gastronomic culture through chocolate. In addition, chocolate products, cocoa beans, and chocolate compositions are exhibited, along with chocolate-themed events such as chocolate fashion shows, live-music performances by singers from cocoa producing countries, and chocolate dessert making competitions [12]. The Salon du Chocolat was held for three days from 10 January to 12 January 2020, celebrating its sixth anniversary in Seoul. A total of 124 companies from 13 countries participated in the exposition and presented information on the trends in their markets, to approximately 40,000 visitors [13].

Schmitt [14] conceptualized five types of holistic experiences, i.e., sense, feel, think, act, and relate, and referred to them as strategic experience modules (SEMs) [9]. Experiences can promote positive perception and willingness to participate among consumers and eventually improve their satisfaction and loyalty by stimulating change of perception and new ideas through learning opportunities [15]. A strong positive correlation exists between the influence of experiential factors and consumers' perceived value [14,16].

Customer delight refers to a profoundly positive emotional state elicited after a feeling of satisfaction and is characterized by strong joy and surprise [17]. This emotion enhances the customer's intention to repurchase core services [18] and is also connected with potential corporate loyalty [19]. Thus, customer delight and satisfaction are a necessary condition for long-term business success [20].

Hoch and Deighton [21] noted that the information obtained from an individual's past experiences was related to the motivation to purchase. They discussed the importance of memory in imparting relevance to the product and stated that experiences stored in an individual's memory were perceived as reliable. Recollection, considered to be positive memory that reminds individuals of their experiences, actions, and events, has been used as an intervention technique to help satisfy the needs and desires of an individual [22]. Accordingly, marketing studies have validated a positive correlation between memorized experience and future decision making [23].

The analysis of the experiential factors influencing the cognitive process of visitors to the chocolate exposition, and the investigation of the association of these factors with post-visit recollection, including delight and memory continuity, are crucial for the success of international chocolate expositions and for boosting the revisit intention of visitors. Such attempts will also render international expositions as differentiated, functional, and sustainable. This study aims to examine the relationship among holistic experiences (SEMs) of the Salon du Chocolat Seoul visitors and their delight, recollection, and revisit intention. The results analyze the impact of key antecedents of the behavioral intentions of visitors to future chocolate expositions. This study provides important practical implications for the content development and marketing strategies required to achieve a continuous flow of visitors to such expositions.

## 2. Literature Review and Hypotheses

### 2.1. Experience, Delight, and Recollection

Experience can broadly be defined as the experience itself and post-experience recollection and understanding [24], and includes all the activities (seeing, hearing, feeling, thinking, and interacting with others) that are performed by individuals [9]. Experience refers to a personal event resulting from direct observation or participation in a specific context, in real life or a virtual setting, or dream [25].

This allows consumers to immerse themselves in a product, service, or brand [26]. Hence, experience is a long-term and continuous process, rather than a short-term outcome and should be understood as an overall perception formed through the stimulating effects of all the brand-related factors [27].

As compared with traditional marketing which focuses on functional aspects, experiential marketing emphasizes the consumer's experience (CX) of consuming products and services [14]. Consumers are deemed to find more value when participating in experiential settings created by experience providers [28]. Owing to this, businesses and brand owners need to utilize holistic CXs that contain rational and emotional factors, as a multifaceted marketing strategy and tool under the experiential marketing approach [14]. In this regard, Schmitt [14] classified the types of CX into "sense," "feel," "think," "act", and "relate" and referred to them as SEMs, addressing emotional domains; this attempt indicated a departure from conventional empirical research that focused on rational aspects. In his unconventional empirical research, Schmitt claimed that the five major elements of SEMs must be understood from a holistic perspective; he asserted that when such holistic CXs were created, customer loyalty toward the brand and brand value improved [9]. Experiential marketing is intended to enhance consumers' memories or knowledge through emotional or unique experiences, such as a feeling of playfulness, during a series of interactions with experiential elements [29]. Experiential marketing seeks to provide memorable and unique experiences for consumers who are considered rational and emotional, with a particular interest in obtaining pleasurable experiences [29,30].

Mehrabian and Russell [31] stressed that the human emotion of delight reflected the degree of positivity in reacting to an experience and connected it to happiness, satisfaction, and love. From a business perspective, consumer delight is a vital factor because it makes consumers stay longer in the store [32] and leads to more purchases [33]. Delight is defined as a positive or emotional response that reflects exciting moods such as pleasure, liking, happiness, exuberance, and fun, all of which are hedonistic emotions elicited by stimulating experiences [34]. Researchers such as Finn [19] and Torres and Kline [35] described delight as an emotion combining high levels of pleasure (joy, elation) and arousal [36]. Customer delight is an emotion comprised of joy and surprise under the affect-based approach [37] and refers to an emotional reaction triggered by pleasant surprise and positive outcomes, that is, the emotional state of being delighted represents a mixture of happiness and surprise [37,38].

Previous studies have reported a positive causal relationship between CX and customer delight in the hospitality industry, given that delight is derived from experience with numerous market stimuli, provided by companies and brands [28,38,39]. Hence, experience-led delight is one of the indicators of a change in the consumer's psychological elements. This type of emotional effect is deemed to be an important factor in creating a halo effect; emotional delight positively influences the overall evaluation of a product after the consumers' perception of their experience [40].

Creating positive memories for tourists/people is essential in the tourism and hospitality industry as these industries provide intangible experiential goods and services, which become more meaningful and important when remembered by the consumers [41]. The recollection of unique experiences is a basic feature of autobiographical memories in certain social and cultural settings [42]. Ochsner [43] defined recollection as, "a process that brings back details specific to a given episode" [44]. Recollection is an individual's habit of thinking about their particular past experiences [45] and the ability to store and, subsequently, recall past activities and experiences consciously [46]. Additionally, recollection refers to the mental process in which past events and situations are recalled in a multidimensional and natural way [47]. Recalling past individual and social experiences also leads people to organize and integrate their intellectual and emotional experiences [48].

Morgan and Xu [49] reported that recollection played the role of extending experiences even when time elapsed after each experience. They also reported that recollection was a potential factor influencing a consumer's impression of experience providers when sharing experiences with others. Kang et al. [44] emphasized that a combination of experiential elements and entertainment on a cruise ship would influence positive recollection by passengers. Thus, previous research on tourists' recollection has underlined the importance of marketing strategies designed to promote positive and

memorable CXs in the hospitality industry [50]. Recollection is found to have a positive effect on the future behavior involved in recalling past positive experiences [22]. The aforementioned studies supported the claim that a holistic experience, created based on Schmitt's SEMs [14], was a positive factor influencing consumer delight and recollection. On the basis of these previous studies, we developed the following hypotheses:

**Hypothesis 1 (H1).** *The holistic experience of the chocolate exposition has a positive impact on delight.*

**Hypothesis 2 (H2).** *The holistic experience of the chocolate exposition has a positive impact on recollection.*

### 2.2. Delight and Recollection

Oh, Fiore, and Jeoung [51] defined experiences as events that evoked positive or negative emotional responses, which in turn facilitated the recollection process. Such emotion denotes a mental state that arises from consumers' own perceptions and evaluations of an event [52]. The pleasure and arousal triggered by experiences can be regarded as the positive effects of a memorable experience [51]. Tung and Ritchi [53] argued that experiencing pleasant or even hedonistic feelings led to positive memories and activities. In addition, Levin and Edelstein [54] stated that positive emotions experienced by consumers while shopping had a positive impact on their memory and future purchasing behavior. Pine and Gilmore [55] asserted that the "sweet spot", a term for ideal consumer experiences, could be used as a means of creating unforgettable memories, beyond the mere delivery of products and services. Accordingly, to develop something memorable and eventually create genuine CX, products and services can be engineered/developed with the goal of creating a personalized and pleasant experience for each consumer [56]. Delight boosts positive emotions of the participants involved in the experiences, thereby encouraging their immersion in activities and improving value-based recognition [51]. This suggests that experiences that evoke positive emotions (e.g., joy and excitement) are well-stored in memory [50]. Additionally, messages that are interesting and evoke pleasant feelings are more memorable than the opposite, i.e., positive messages are more unforgettable than negative messages [57]. According to these findings, we developed the following hypothesis:

**Hypothesis 3 (H3).** *The delight of the chocolate exposition has a positive impact on recollection.*

### 2.3. Delight, Recollection, and Revisit Intention

Consumers' revisit intention is a key factor in determining the outcome of relationship marketing, as it is useful for predicting a consumer's future behavioral intention toward a continuing relationship [58]. Revisit intention refers to the consumer's willingness to use previously identified products/services and/or stores [59]. In other words, revisit intention is a response to the predicted effect of the future transaction that could ensue if the consumer is satisfied with the services [60].

Previous studies have reported that positive consumption elicited emotions leading to satisfaction, recommendation, and revisits [38,61–63]. Oliver [61] and Bigné and Andreu [63] noted that there was a positive correlation between consumption that elicited positive emotion and satisfaction, which led, first, to a positive attitude, and progressed to pleasant experiences, loyalty, and behavioral or revisiting intention. According to Bloemer and de Ruyter [62], positive emotions (e.g., pleasant feelings) exerted a positive effect on some components of loyalty such as, behavioral intention and willingness to pay. In summary, positive arousal and delight are positively correlated with satisfaction and more importantly, with the intention to repurchase (revisit intention) [61].

As consumers tend to pursue more attractive, unique, and memorable experiences in certain situations [64], experiential factors influence their emotional attachment and loyalty (e.g., revisit, recommendation, and positive word of mouth) through memory [65].

That is, memory affects future behavior according to recall-based assessment [66]. Kang et al. [44] found that recalling cruise travel had a positive effect on behavioral intention to recommend and revisit. According to Tussyadiah, Park, and Fesenmaier [67], memories of previous travel experiences increased destination knowledge and ultimately led to revisit intention. Therefore, well-organized experiences with positive memories influence overall satisfaction and consideration of future behavior such as the intention to revisit and recommend [51]. Thus, recollection of past experiences impact future behaviors such as revisit intention, based on high reliability [68]. On the basis of these previous studies, we developed the following hypotheses:

**Hypothesis 4 (H4).** *The delight of the chocolate exposition has a positive impact on revisit intention.*

**Hypothesis 5 (H5).** *The recollection of the chocolate exposition has a positive impact on revisit intention.*

The model development was based on previous literature and included a synthesis of their proposals into practical guidelines for researchers [69]. Researchers arrived at the final model through extensive discussions, and unanimous agreement on all its constructs and dimensions. Figure 1 depicts the research model containing all the hypotheses.

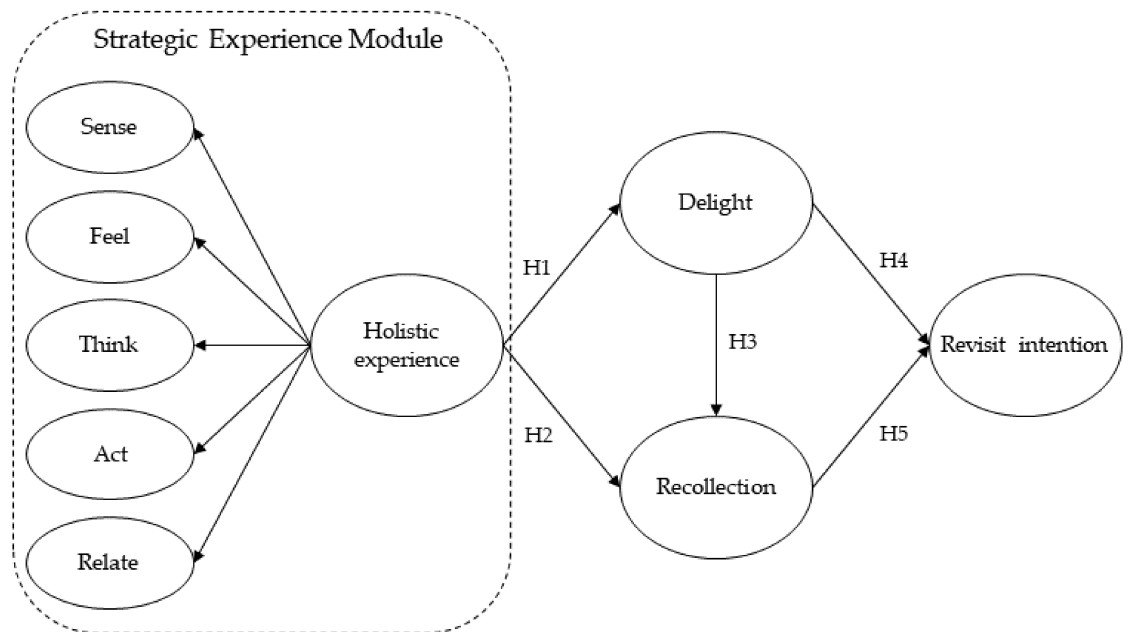

**Figure 1.** Research model.

## 3. Methodology

### 3.1. Sampling and Data Collection

This study included South Korean residents (male and female) above the age of 13 who visited the chocolate exposition, Salon du Chocolat Seoul (refer Table 1). Data were collected over 10 days using online surveys from 15–24 July 2019, by Macromill Embrain, which is a leading, reliable online research company in Korea, with a panel of more than 1.3 million consumers. Screening questions were administered before a respondent was invited for an interview. Potential participants were asked if they had ever visited the chocolate exposition; those who had visited the exposition were briefed on the objective and scope of the study. Their consent was obtained before initiating study procedures. The questions were presented to all respondents in the same order. A sample of 280 respondents was selected and used for the empirical analysis. A profile of the sample is shown in Table 1. Since the net response rate was 4.8%, the dataset was subjected to wave analysis, which is a

non-response bias analysis. Answers gathered in the initial five days were classified as early responses, while those gathered during the last five days were classified as late responses. Using both groups, an independent t-test was carried out and the results indicated a non-significant difference between them, thereby indicating the absence of a non-response bias within the dataset.

**Table 1.** Respondents' profiles.

| Demographic Characteristics | | Frequency | Percentage |
|---|---|---|---|
| Gender | Male | 58 | 20.7 |
| | Female | 222 | 79.3 |
| | 13–19 years | 49 | 17.5 |
| | 20–29 years | 187 | 66.8 |
| Age | 30–39 years | 27 | 9.6 |
| | 40–49 years | 10 | 3.6 |
| | Above 50 years | 7 | 2.5 |
| | Middle and high school student | 57 | 20.4 |
| | University student | 90 | 32.1 |
| | Sales and service | 48 | 17.1 |
| | Office workers | 16 | 5.7 |
| Occupation | Technician | 7 | 2.5 |
| | Professional job | 34 | 12.1 |
| | Housewife | 9 | 3.2 |
| | Others | 19 | 6.8 |
| | 1–2 times | 130 | 46.4 |
| Frequency of purchase for 1 month | 3–4 times | 86 | 30.7 |
| | Above 5 times | 64 | 22.9 |
| | 1 time | 215 | 76.8 |
| Frequency of visits to the | 2 times | 44 | 15.7 |
| exposition | 3 times | 17 | 6.1 |
| | Above 4 times | 4 | 1.4 |
| | Chocolate fashion show | 60 | 21.4 |
| | Dynamic performance | 11 | 3.9 |
| | Chocolate recipe demonstration | 92 | 32.9 |
| Favorite exposition program | Experiential learning about chocolate | 67 | 23.9 |
| | Seminar with experts | 24 | 8.6 |
| | Competition for chocolate national representatives | 11 | 3.9 |
| | Others | 15 | 5.4 |

## 3.2. Research Instruments

This study organized the survey questions from a literature review and modified them to suit the context of a chocolate exposition. After drafting the initial questionnaire, three professors of restaurant management and two chocolate exposition operators were consulted on its composition and contents. Prior to the survey, a pilot test was conducted to determine whether the participants fully understood the survey questions. On the basis of the pilot test, questions that were difficult to answer accurately were removed and the remaining questions were modified as required.

This study consisted of the following eight factors: delight; recollection; revisit intention; and the five factors of sense, feel, think, act, and relate, which form the constructs of holistic experience (see Table A1). The five factors in holistic experience each consisted of three items, resulting in a total of fifteen items, which were derived from Ding and Tseng [70], Schmitt [14], and Yoo et al. [9]. Delight was composed of five items, obtained from Ali et al. and Ma et al. [28,36]. Recollection was composed of three items adopted from Manthioua et al. [60]. Revisit intention was composed of three items as well, obtained from Yoo et al. [9]. All scale items were measured on a five-point Likert-type scale ranging from "strongly disagree" to "strongly agree."

*3.3. Analytical Methods*

Our analysis used the Statistical Package for the Social Sciences (SPSS) 22.0 and Analysis of a Moment Structures (AMOS) 22.0. The demographic characteristics were analyzed using SPSS 22.0. Data analysis was carried out using Anderson and Gerbing's [71] two-step approach, i.e., measurement model and structural model evaluation, to test our hypotheses. Confirmatory factor analysis (CFA) was first conducted to test the adequacy of the measurement model and assess composite reliability and convergent validity. Specifically, experience was measured with the second-order CFA model. Then, structural equation modeling (SEM) was performed to test for hypothetical relationships among the four constructs proposed in the conceptual model.

## 4. Data Analysis and Results

*4.1. Measurement Model*

The goodness-of-fit of the measurement model was assessed using CFA, according to the cut-off values of the following seven fit indices: $\chi^2$/df (<3), goodness-of-fit index (GFI > 0.90), root mean square error of approximation (RMSEA < 0.08), root mean square residual (RMR < 0.08), normed fit index (NFI > 0.9), incremental fit index (>0.9), and comparative fit index (CFI > 0.9) [71]. The measurement model had a good fit with the data ($\chi^2$ = 149.882, df = 88, p= 0.000, CMIN/df = 1.703, RMR = 0.034, GFI = 0.938, NFI = 0.943, IFI = 0.976, CFI = 0.975, and RMSEA = 0.050). The adequacy of the measurement model was tested using the standard criteria of reliability, convergent, and discriminant validity. First, reliability was assessed based on composite construct reliability values. As shown in Table 2, all the values exceeded 0.7, demonstrating adequate composite reliability [72]. The average variance extracted (AVE) values of all the constructs were higher than the minimum threshold of 0.5, indicating the convergent validity of the measures [72].

**Table 2.** Reliability and confirmatory factor analysis for measurement items.

| Variables & Items | Standardized Loading | CCR [a] | AVE [b] |
|---|---|---|---|
| **Experience ($\alpha$ = 0.850)** | | | |
| Sense | 0.554 | 0.841 | 0.520 |
| Feel | 0.646 | | |
| Think | 0.764 | | |
| Act | 0.778 | | |
| Relate | 0.506 | | |
| **Delight (DL) ($\alpha$ = 0.871)** | | | |
| DL1 | 0.774 | | |
| DL2 | 0.715 | 0.888 | 0.634 |
| DL3 | 0.741 | | |
| DL4 | 0.794 | | |
| DL5 | 0.719 | | |
| **Recollection (RC) ($\alpha$ = 0.849)** | | | |
| RC1 | 0.773 | 0.858 | 0.669 |
| RC2 | 0.817 | | |
| RC3 | 0.848 | | |
| **Revisit intention (RI) ($\alpha$ = 0.884)** | | | |
| RI1 | 0.865 | 0.888 | 0.724 |
| RI2 | 0.842 | | |
| RI3 | 0.838 | | |

Note: [a], composite construct reliability and [b], average variance extracted.

To examine the discriminant validity of those variables with verified convergent validity, we compared the square root of AVE of each latent variable against the corresponding correlation coefficients between potential variables. Table 3 shows that the square root of AVE of each latent

variable is larger than its corresponding correlation coefficient, indicating adequate discriminant validity [73].

**Table 3.** Correlations of analysis between the variables.

| Variable | 1 | 2 | 3 | 4 |
|---|---|---|---|---|
| 1. Experience | **0.721** | | | |
| 2. Delight | 0.638 | **0.796** | | |
| 3. Recollection | 0.536 | 0.525 | **0.818** | |
| 4. Revisit intention | 0.618 | 0.524 | 0.495 | **0.851** |
| Mean | 3.615 | 3.962 | 3.503 | 3.797 |
| S.D. | 0.594 | 0.699 | 0.858 | 0.891 |

Note: Diagonal elements show the square root of AVE. Below the diagonal is the corresponding correlation coefficient. All correlation coefficients were significant at the 0.01 level.

### 4.2. Structural Model

SEM was conducted using the AMOS 22.0 statistical package. To test the hypotheses established through the SEM path coefficients, the fit of the structural model describing the relationships among the constructs was assessed. The model fit indices were $\chi^2$ = 228.301, df = 94, $p$ = 0.000, CMIN/df = 2.429, RMR = 0.056, GFI = 0.912, NFI = 0.913, IFI = 0.947, CFI = 0.946, and RMSEA = 0.072, which met the standard assessment criteria. The result of each hypothesis test describing the causal relationship between pairs of constructs is presented in Figure 2. H1 was supported indicating that holistic experience positively and significantly influenced delight ($\beta$ = 0.780, t = 6.965, $p$ = 0.000). H2 was supported indicating that holistic experience positively and significantly influenced recollection ($\beta$ = 0.310, t = 2.668, $p$ = 0.008). H3 was supported indicating that delight positively and significantly influenced recollection ($\beta$ = 0.367, t = 3.242, $p$ = 0.001). H4 was supported because delight positively and significantly influenced revisit intention ($\beta$ = 0.440, t = 5.589, $p$ = 0.000). Lastly, H5 was supported because recollection positively and significantly influenced revisit intention ($\beta$ = 0.293, t = 3.794, $p$ = 0.000).

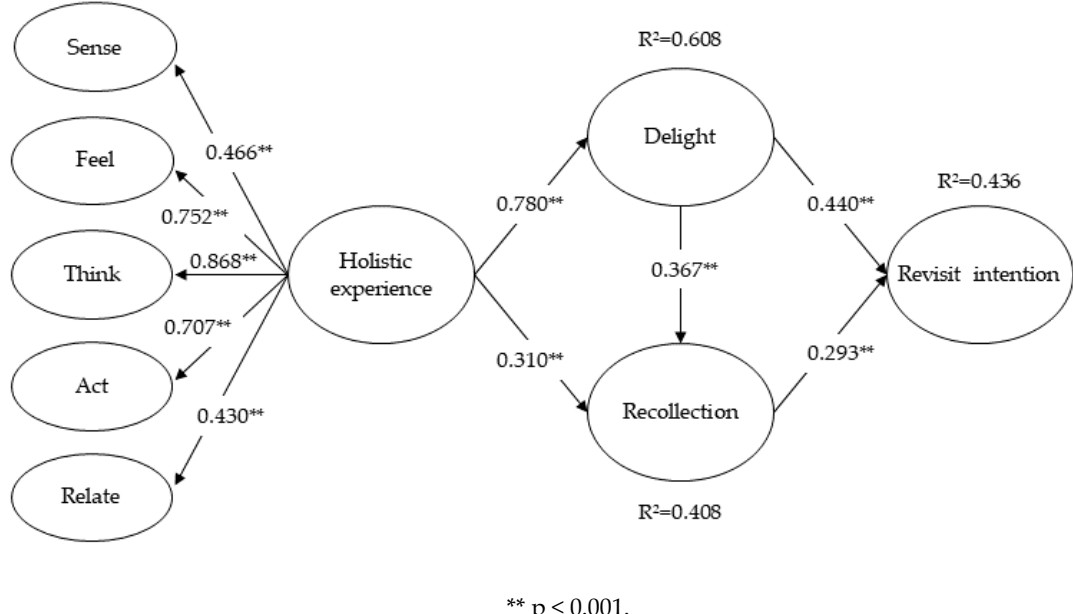

** $p$ < 0.001.

**Figure 2.** Structural equation model with parameter estimates.

## 5. Discussion and Conclusions

### 5.1. Theoretical Implications

From a theoretical perspective, this study aimed at confirming the antecedent variables influencing revisit intention in visitors to the chocolate exposition. We applied the SEMs conceptualized by Schmitt [14], which have reaped attention as a theoretical concept for enhancing brand loyalty and creating brand value. Although research on food-oriented festivals and tourism has been actively conducted [5,6,74–76], few studies have focused on food expositions. Additionally, food-related festivals and tourism studies [5,6,74–76] have primarily addressed the participants' motivations and examined choice attributes as focal variables, suggesting the need for further studies to explore participants' experiences. Given this, the present study applied experiential marketing theory for visitors at the chocolate exposition through a novel approach. Additionally, we presented a model that included customer delight and recollection, which have been individually validated by previous empirical studies on the hospitality industry [36,38,44]. Thus, this study design differs from previous empirical research on the tourism and hospitality industry. As an empirical research, this is the first study involving visitors at a chocolate exposition in the tourism and hospitality industry. Therefore, this study contributes to the literature on CX by applying an experiential marketing theory to the context of an exposition.

The results revealed that experiential marketing used for Salon du Chocolat Seoul visitors effectively influenced the visitors' delight and recollection and induced revisit intention. Delight was found to be an important variable in inducing positive recollections through a subjective assessment of their experiences in the exposition. This suggests that delight positively influences recollection, which is related to long-term memories and revisit intention, thereby demonstrating the effect of the leading variable. Finally, this study applied Schmitt's SEMs [14] of delight and recollection as the leading variables associated with behavioral intention to revisit the chocolate exposition, confirmed the structural relationship between these variables, and validated the adequacy of the model. Thus, the study design and results significantly contribute to the theory concerning the exposition for food products.

### 5.2. Practical Implications

Eatertainment, defined as a consumption behavior combining eating with entertainment, has become evident [8]. It is widely experienced with various forms of food, including local food festivals and tours [5]. Recently, the dessert sector, in Korea has seen an increasing shift toward fun-oriented consumption, particularly among millennial consumers, and has become the axis of the food service industry [77]. In line with this consumption trend, the various experiential elements offered by the chocolate exposition were meaningful, because of their ability to satisfy the eatertainment needs of visitors. The identification of visitors' experiences in the exposition is a necessary part of the marketing strategy to appeal to the senses and create long-term memories.

Chocolate expositions need to offer a variety of exciting experience-based programs, such as cooking classes, chocolate-making using characters, and DIY kits, that visitors can participate in with their families and companions. The cocoa aroma in chocolate is said to provide emotional comfort [78]. Therefore, experiencing therapeutic benefits by using cocoa soaps, cosmetics, or fragrances/aroma should lead to emotional relief and mental stability. Furthermore, stakeholders involved in dessert and chocolate manufacturing can provide information on the various foods, cooking tips, and technical know-how to aid consumer understanding. Thus, consumers may develop a sharper positive perception after an exposition, with increasing loyalty to the event.

In the hospitality industry, good facilities and a clean service environment serve as a source of joy among users [17]. Therefore, it is important to select a venue with due consideration to the physical environment including the venue facilities, parking areas, and accessibility. Similarly, the contact point service provided to customers constitutes an important element in customer delight [36].

The staff members' professional knowledge and services may be remembered as a differentiated experience and, subsequently, inculcate pleasant feelings [35]. Staff members working at the site need to be trained before operating booths, managing events, and meeting the needs of visitors.

Impressive experiences reinforce recollection [55], while technological experiences enhance the level of recollection [79]. This means that food-related festivals, expositions, and the convention industry need to break away from the traditional approach and explore hyper-connectivity and superintelligence in tandem with the fourth industrial revolution. If a chocolate digitech experience zone and package booths are prepared in an exposition using the latest augmented reality (AR), virtual reality (VR), and artificial intelligence (AI) technologies, the visitor experience should be more stimulating and impressive, leading to favorable post-visit evaluations.

Online platforms (e.g., Web, App, and YouTube) that showcase expositions are the same media used to deliver entertainment to customers [17] and require continuous management. To adapt to rapidly changing trends in the untact era, increased opportunities for online digital activities (e.g., short experiences before visiting, streaming, and LAN tours) can be provided to meet customer needs. In particular, it is necessary to provide a chatbot service on social networks, which can be easily accessed by customers and used to disseminate information and respond to customer/service inquiries in accordance with customer preferences. Chatbots are poised to transform customer service activity over the next decade. Essentially, chatbots are software algorithms capable of interacting with humans; they use big data analytics and technologies such as natural language processing and machine learning to develop accurate profiles of users and interact with them [80]. Developing short-form promotional video content aimed at Generation Z consumers is also useful and follows the paradigm of the digital content market.

Finally, if convening the exposition becomes uncertain due to pandemic-related safety requirements, the organizers of the exposition should actively utilize AR and VR technologies. The study by Redondo et al. [81] strategically modified the method of participation in events and expositions. They applied gamification for potential visitors to international expositions using VR. They tested a new interaction methodology by joining touch screens with large-sized viewer screens to create a semi-immersive environment. The participation of citizens in their project has increased exponentially, thus enabling the validation of its agility with different people engaging with it simultaneously.

Likewise, if the chocolate exposition cannot be held in a physical form, as before the pandemic, it will be necessary to create virtual exposition content. Participants should be guided to a virtual showroom. The virtual showrooms should have a wider variety of content than offline expositions and should simulate a delightful and authentic experience for the participants. In addition, experiential programs such as chocolate making can be performed through real-time online streaming. Chocolate ingredients are delivered to the preregistrants of the exposition, and the instructor's demonstration and explanation of chocolate making, question and answer sessions, and coaching for participants is conducted in real time through a video system (e.g., ZOOM). Untact experiences such as these would be a progressive way to prepare for the post-COVID era.

### 5.3. Conclusions

This study examined how customer delight, recollection, and revisit intention were related to their experiences in the chocolate exposition. We presented a model including the holistic experience of SEMs, delight, recollection, and revisit intention. The results revealed that experience was a positive determinant of customer delight and recollection; and these delight-related factors rated as higher emotional experiences than satisfaction, and positively influenced recollection and revisit intention. These empirical findings prove that the model proposed in this study is adequate for explaining experiences in the chocolate exposition and the visitors' reactions and behaviors. Hence, the theoretical framework used for model validation could serve as the basis for further studies on food exposition research.

*5.4. Limitations and Future Research*

Despite the results and implications of the study, there are some limitations. First, the results of this study may not be generalizable because data was collected in Korea only, although the Salon du Chocolat Seoul is an international exposition. Given that the chocolate exposition is still in its early stages in Korea, the application of the findings to developed countries with higher chocolate consumption is inappropriate. Future studies are necessary to predict consumption behaviors in relation to chocolate by comparing the differences in consumption characteristics among countries having different consumption levels (high, medium, and low). Second, the analysis of the differences between groups divided according to the frequency of visits will also facilitate an understanding of visitor behavior and the development of action plans for experiential marketing planning and strategy.

**Author Contributions:** H.M.J. and S.H.K. conceived and designed the experiments; H.M.J. performed the experiments and analyzed the data; H.M.J., S.R.Y., and S.H.K. wrote the paper. All authors have read and agreed to the published version of the manuscript.

**Funding:** This research received no external funding.

**Conflicts of Interest:** The authors declare no conflict of interest.

## Appendix A

**Table A1.** Measurement items for study.

| | Measurement Items |
|---|---|
| | **Sensory** |
| SE1 | This chocolate exposition tries to engage my senses. |
| SE2 | This chocolate exposition is perceptually interesting. |
| SE3 | This chocolate exposition has a sensory appeal. |
| | **Feel** |
| FE1 | This chocolate exposition tries to put me in a certain mood. |
| FE2 | This chocolate exposition makes me respond in an emotional manner. |
| FE3 | This chocolate exposition appeals to feelings. |
| | **Think** |
| TH1 | This chocolate exposition tries to intrigue me. |
| TH2 | This chocolate exposition stimulates my curiosity. |
| TH3 | This chocolate exposition appeals to my creative thinking. |
| | **Act** |
| AC1 | This chocolate exposition makes me think about my lifestyle. |
| AC2 | This chocolate exposition reminds me of activities that I can do. |
| AC3 | This chocolate exposition leads me to think about my actions and behaviors. |
| | **Relate** |
| RE1 | This chocolate exposition tries to get me to think about relationship. |
| RE2 | I can relate to other people through this chocolate exposition. |
| RE3 | This chocolate exposition reminds me of social rules and arrangements. |
| | **Delight** |
| DL1 | I felt delighted during my visit. |
| DL2 | I felt gleeful during my visit. |
| DL3 | I felt elated during my visit. |
| DL4 | I felt enthusiastic during the visit. |
| DL5 | I felt excited during the visit. |
| | **Recollection** |
| RC1 | When I remember this chocolate exposition, I feel as though I am reliving the original chocolate exposition experience. |
| RC2 | I can actually remember this chocolate exposition instead of just knowing it happened. |
| RC3 | While remembering this chocolate exposition experience, I feel like I traveled back to the time it happened. |
| | **Revisit intention** |
| RI1 | I intend to revisit this chocolate exposition next time. |
| RI2 | I plan to revisit this chocolate exposition continuously. |
| RI3 | It is very likely that I will revisit this chocolate exposition. |

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
