# Peer review of "The Relationships among Experience, Delight, and Recollection for Revisit Intention in Chocolate Exposition"

_sustainability, doi:10.3390/su12208644_

Round 1
Reviewer 1 Report
Thank you for inviting me to review this paper. The paper examines the relationships between four constructs related to visiting a chocolate exposition/exhibition. I have recommendations for improvement.
Before I start providing my recommendations I need to acknowledge that I used to use the survey method in my research, but I have not used it in the last few years and my recent research focuses on the case study method. My comments are therefore focusing on the positioning of the work (not the statistical parts) and the practical implications of the paper.
The abstract says: “this study examined the relationship among holistic experience and delight, recollection and revisit intention in visitors to chocolate exposition to identify factors influencing their revisit intention.”. How is it possible that someone can identify factors from relationships among factors? I’ve read the paper and did not find any new factor. The study is only trying to examine and confirm the relationships among factors. I would suggest that the authors clarify the purpose of their research, because if they claim that they have identified factors/antecedents I (as a reviewer and reader in general) expect to see those newly identified factors/antecedents. If there is no new factor identified (which is the case in your study) please don’t claim that the study has identified factors. In my opinion, it the study is examining/confirming the relationships among factors. Please correct this in the entire paper (abstract, introduction, discussion...).
In the abstract and the main text: expositions or exhibitions? I would suggest that you choose the most correct/relevant word and use it consistently in the abstract and main text.
This sentence sounds wrong: “This study consists of eight factors: the sense, feel, think, act, and relate factors which form the basis of the constructs of holistic experience, which are delight, recollection, and revisit intention.”. Please correct it.
In terms of methodology for developing the model, I don’t understand how the authors approached the factors (the constructs in the model and dimensions of the holistic experience construct). Did the authors just reuse these factors from literature without any discussion among themselves or did they have some discussion/brainstorming? Did all authors (the research team) agree on using all of those factors or for example one of the authors was not fully agreeing? A couple of sentences that clearly mention how the authors discussed the factors and whether they all agreed on all constructs and dimensions need to be provided. I would suggest reading and using this very recent and relevant reference that provides guidelines on reporting author agreement on factors:
Nili, A., Tate, M., Barros, A., & Johnstone, D. (2020). An approach for selecting and using a method of inter-coder reliability in information management research. International Journal of Information Management, 54, 102154.
Given that the paper does not include much novelty in terms of theory, I was expecting to see a good amount of specific information on practical contribution/recommendations (which the paper is not sufficiently providing). I have a few suggestions for improvement: first, I don’t understand this sentence: “Entertainment, defined as a consumption behavior combining eating with entertainment, has become evident”. Please correct it. Second, the authors mention: “our study also proposes practical suggestions for hosting food expositions in a post-COVID world”. How? I’m struggling with seeing this in the paper. Currently the paper provides little information on this. In terms of practical contribution, as a researcher whose research focuses on the role of technology in consumer/user behaviour, naturally I would suggest writing more about these two areas: (1) “virtual reality and gamification” (please see the first reference below) for conducting the exposition/exhibition, and (2) “chatbots” (please see the second reference below) for the marketing aspect and responding to customer/service enquiries. Please use these recent and relevant references (and other recent ‘and’ relevant references that you find in literature):
Redondo, E., Zapata, H., Navarro, I., Fonseca, D., Gimenez, L., Pérez, M. Á., & Sánchez-Sepúlveda, M. (2020, July). GAME4CITY. Gamification for Citizens Through the Use of Virtual Reality Made Available to the Masses. Viability Study in Two Public Events. In International Conference on Human-Computer Interaction (pp. 315-332). Springer, Cham.
Nili, A., Barros, A., & Tate, M. (2019). The public sector can teach us a lot about digitizing customer service. MIT Sloan Management Review, 60(2), 84-87.
All the best with your revision efforts.
Author Response
Response to Reviewer 1 Comments
Point 1: The abstract says: “this study examined the relationship among holistic experience and delight, recollection and revisit intention in visitors to chocolate exposition to identify factors influencing their revisit intention.”. How is it possible that someone can identify factors from relationships among factors? I’ve read the paper and did not find any new factor. The study is only trying to examine and confirm the relationships among factors. I would suggest that the authors clarify the purpose of their research, because if they claim that they have identified factors/antecedents I (as a reviewer and reader in general) expect to see those newly identified factors/antecedents. If there is no new factor identified (which is the case in your study) please don’t claim that the study has identified factors. In my opinion, it the study is examining/confirming the relationships among factors. Please correct this in the entire paper (abstract, introduction, discussion...).
Response 1: I sincerely appreciate your comment. We agree with your assessment and have amended the text accordingly. Specifically, we have replaced “identify” with “confirm” in the abstract as per your suggestion (see line 13).
Point 2: In the abstract and the main text: expositions or exhibitions? I would suggest that you choose the most correct/relevant word and use it consistently in the abstract and main text.
Response 2: Thank you for highlighting the inconsistent terminology. As per your suggestion, I have standardized the text and replaced all mentions of “exhibition” with “exposition” throughout the revised manuscript.
Point 3: This sentence sounds wrong: “This study consists of eight factors: the sense, feel, think, act, and relate factors which form the basis of the constructs of holistic experience, which are delight, recollection, and revisit intention.”. Please correct it.
Response 3: I sincerely appreciate your comment. I have amended the highlighted sentence in the revised manuscript for clarity (see lines 230-232). For your ease of reference, this sentence has been restructured and amended to, “This study consists of eight factors: delight; recollection; revisit intention; and the five factors of sense, feel, think, act, and relate, which form the constructs of holistic experience.”
Point 4: In terms of methodology for developing the model, I don’t understand how the authors approached the factors (the constructs in the model and dimensions of the holistic experience construct). Did the authors just reuse these factors from literature without any discussion among themselves or did they have some discussion/brainstorming? Did all authors (the research team) agree on using all of those factors or for example one of the authors was not fully agreeing? A couple of sentences that clearly mention how the authors discussed the factors and whether they all agreed on all constructs and dimensions need to be provided. I would suggest reading and using this very recent and relevant reference that provides guidelines on reporting author agreement on factors:
Response 4: I sincerely appreciate your comment. As suggested, I have added a sentence to explain the approach to model development (see lines 201-202). Extracted here: “The model was developed through discussions among the researchers based on the literature review; the researchers unanimously agreed on all constructs and dimensions.”
I hope this adequately addresses your comment.
Point 5: I have a few suggestions for improvement: first, I don’t understand this sentence: “Entertainment, defined as a consumption behavior combining eating with entertainment, has become evident”. Please correct it. Second, the authors mention: “our study also proposes practical suggestions for hosting food expositions in a post-COVID world”. How? I’m struggling with seeing this in the paper. Currently the paper provides little information on this. In terms of practical contribution, as a researcher whose research focuses on the role of technology in consumer/user behaviour, naturally I would suggest writing more about these two areas: (1) “virtual reality and gamification” (please see the first reference below) for conducting the exposition/exhibition, and (2) “chatbots” (please see the second reference below) for the marketing aspect and responding to customer/service enquiries. Please use these recent and relevant references (and other recent ‘and’ relevant references that you find in literature):
Response 5: I sincerely appreciate your suggestion, which aims to improve the quality of this paper. The word “entertainment” was a typographical error; I have corrected it to “eatertainment,” which is explained by the definition that follows.
Regarding your second comment, by a post-COVID world, we meant when it is safe for such food expositions to resume. I have also added content about chatbot services (see lines 348-350). Extracted here: “In particular, it is necessary to provide a chatbot service on social networks, which can be easily accessed by customers and used to disseminate information and respond to customer/service enquiries in accordance with customer preferences.”
Please also refer to lines 340-343 for applications of virtual reality and artificial intelligence in this space. We hope this adequately addresses your comment.

Reviewer 2 Report
Authors make an extensive review of basic terms and concepts in the introduction, hypothesis section, as well as in methods, and give very little scientific additional value with the experiment. Although they did their best to design and describe the experiment I do not see its scientific contribution. This could be published in a professional magazine as a professional paper, not as a scientific paper in a peer-reviewed Journal indexed in leading databases.
Author Response
Response to Reviewer 2 Comments
Point 1: Authors make an extensive review of basic terms and concepts in the introduction, hypothesis section, as well as in methods, and give very little scientific additional value with the experiment. Although they did their best to design and describe the experiment I do not see its scientific contribution. This could be published in a professional magazine as a professional paper, not as a scientific paper in a peer-reviewed Journal indexed in leading databases.
Response 1: I sincerely appreciate your review and would like to bring your attention to subsection 5.1 of the manuscript where we have aimed to highlight the theoretical implications of this study.
Extracted here for ease of reference: “Although research on food-oriented festivals and tourism has been actively conducted [6,73,74,75,5], few studies on focus on food expositions. Additionally, food-related festivals and tourism studies [6,73,74,75,5] primarily addressed the participants’ motivations and examined choice attributes as focal variables, suggesting the need for further studies to explore participants’ experiences. Given this, the present study applied experiential marketing theory to visitors to the chocolate exposition through a novel approach.”
Apart from the approach, we have also extended the application of experiential marketing theory to the context of a food exposition, which has not been done before. Extracted here: “As an empirical research, this is the first study involving visitors to a chocolate exposition in the tourism and hospitality industry. Therefore, this study contributes to the literature on CX by applying experiential marketing theory to the context of an exposition.”
Moreover, we have listed several practical implications of the study results in subsection 5.2.
We hope that our revised paper adequately meets your requirements and is worthy of being published in your esteemed journal.

Reviewer 3 Report
Τhe project was very well executed with a very interesting approach.
Author Response
Response to Reviewer 3 Comments
Point 1: Τhe project was very well executed with a very interesting approach.
Response 1: We thank you for your positive feedback and hope that our revised manuscript is suitable for publishing in Sustainability.

Round 2
Reviewer 1 Report
Thanks for sending the paper to me for another round of review. I read the revised version, but I am not completely convinced about the quality of revision. The are two remaining major issues:
First, the paper still lacks novelty in terms of implications for theory. At best, the current version is a bit lower than borderline in terms of acceptability. In my previous review I mentioned: "given that the paper does not include much novelty in terms of theory, I was expecting to see a good amount of specific information on practical contribution/recommendations (which the paper is not sufficiently providing)." The authors have made minimal effort for writing implications for practice. I invite the authors to review more academic research papers (journal and conference papers) and grey literature (professional magazines) to write more recommendations for practice. At least two more paragraphs need to be added.
The second major issue is that while the authors have added some text for improving the methodology (please see my previous review) and some small text to improve the implications for practice (please see my previous review), they have left that text without any reference. I had provided "the most relevant" references (three papers) to improve these parts of the paper, but none of them has been used. Not using references means not enough validity. If the authors refuse to use these most relevant references for those parts of the paper, then they need to explain why (in their response file) and add other highly references for those parts of the paper.
I suggest major revision. I however also believe that revising the paper shouldn't require much time. Good luck with your revision.
Author Response
Response to Reviewer 1 Comments
Point 1: Thanks for sending the paper to me for another round of review. I read the revised version, but I am not completely convinced about the quality of revision. There are two remaining major issues:
First, the paper still lacks novelty in terms of implications for theory. At best, the current version is a bit lower than borderline in terms of acceptability. In my previous review I mentioned: "given that the paper does not include much novelty in terms of theory, I was expecting to see a good amount of specific information on practical contribution/recommendations (which the paper is not sufficiently providing)." The authors have made minimal effort for writing implications for practice. I invite the authors to review more academic research papers (journal and conference papers) and grey literature (professional magazines) to write more recommendations for practice. At least two more paragraphs need to be added.
Response 1: I sincerely appreciate your comment. I reviewed the papers that you recommended and professional magazines, and found them to be valuable resources. We have thus added more practical implications to the revised manuscript. We hope that the additional information will now adequately complete the paper.
Point 2: The second major issue is that while the authors have added some text for improving the methodology (please see my previous review) and some small text to improve the implications for practice (please see my previous review), they have left that text without any reference. I had provided "the most relevant" references (three papers) to improve these parts of the paper, but none of them has been used. Not using references means not enough validity. If the authors refuse to use these most relevant references for those parts of the paper, then they need to explain why (in their response file) and add other highly references for those parts of the paper.
Response 2: I sincerely thank the reviewer for this pertinent point and apologize for the lapse. The relevant papers were reviewed and have now been included in the list of references.

Reviewer 2 Report
Although the manuscript has been corrected according to other reviewer's comments, I still do not see real scientific contribution.
Author Response
Response to Reviewer 2 Comments
Point 1: Although the manuscript has been corrected according to other reviewer's comments, I still do not see real scientific contribution.
Response 1: I sincerely appreciate your review. We hope that our revised paper adequately meets your requirements and is worthy of being published in your esteemed journal.

Round 3
Reviewer 1 Report
I read the parts of the paper that the authors have shown in different colour (the parts that have been revised). The authors have made some acceptable level of effort for revising the paper. I agree with acceptance of the paper if the senior editor agrees too.